# Phenotypic Response to Light Versus Shade Associated with DNA Methylation Changes in Snapdragon Plants (*Antirrhinum majus*)

**DOI:** 10.3390/genes12020227

**Published:** 2021-02-04

**Authors:** Pierick Mouginot, Nelia Luviano Aparicio, Delphine Gourcilleau, Mathieu Latutrie, Sara Marin, Jean-Louis Hemptinne, Christoph Grunau, Benoit Pujol

**Affiliations:** 1PSL Université Paris: EPHE-UPVD-CNRS, USR 3278 CRIOBE, Université de Perpignan, 52 Avenue Paul Alduy, CEDEX 9, 66860 Perpignan, France; pierick.mouginot@univ-perp.fr (P.M.); mathieu.latutrie@univ-perp.fr (M.L.); marin.sa31@gmail.com (S.M.); 2Université Montpellier, CNRS, IFREMER, UPVD, Interactions Hôtes Pathogènes Environnements (IHPE), 66860 Perpignan, France; nelia.luviano@univ-perp.fr (N.L.A.); christoph.grunau@univ-perp.fr (C.G.); 3Laboratoire Évolution & Diversité Biologique (EDB, UMR 5174), Université Fédérale de Toulouse Midi-Pyrénées, CNRS, IRD, UPS, 118 route de Narbonne, Bat 4R1, CEDEX 9, 31062 Toulouse, France; delphine.gourcilleau@gmail.com (D.G.); jean-louis.hemptinne@educagri.fr (J.-L.H.)

**Keywords:** phenotypic plasticity, epigenetics, epiGBS, stem elongation, shade avoidance

## Abstract

The phenotypic plasticity of plants in response to change in their light environment, and in particularly, to shade is a schoolbook example of ecologically relevant phenotypic plasticity with evolutionary adaptive implications. Epigenetic variation is known to potentially underlie plant phenotypic plasticity. Yet, little is known about its role in ecologically and evolutionary relevant mechanisms shaping the diversity of plant populations in nature. Here we used a reference-free reduced representation bisulfite sequencing method for non-model organisms (epiGBS) to investigate changes in DNA methylation patterns across the genome in snapdragon plants (*Antirrhinum majus* L.). We exposed plants to sunlight versus artificially induced shade in four highly inbred lines to exclude genetic confounding effects. Our results showed that phenotypic plasticity in response to light versus shade shaped vegetative traits. They also showed that DNA methylation patterns were modified under light versus shade, with a trend towards global effects over the genome but with large effects found on a restricted portion. We also detected the existence of a correlation between phenotypic and epigenetic variation that neither supported nor rejected its potential role in plasticity. While our findings imply epigenetic changes in response to light versus shade environments in snapdragon plants, whether these changes are directly involved in the phenotypic plastic response of plants remains to be investigated. Our approach contributed to this new finding but illustrates the limits in terms of sample size and statistical power of population epigenetic approaches in non-model organisms. Pushing this boundary will be necessary before the relationship between environmentally induced epigenetic changes and phenotypic plasticity is clarified for ecologically relevant mechanisms with evolutionary implications.

## 1. Introduction

Snapdragon plants (*Antirrhinum majus* L.) undergo developmental changes resulting in different morphologies after exposure to shade [1]. This is one if not the most common example of phenotypic plasticity in plants where changes in internode length (stem elongation), apical dominance (reduced branching), and photosynthetic efficiency (increased Specific Leaf Area or SLA) are observed following shade exposure [1,2,3]. When it allows plants to avoid the presence of neighbouring vegetation, it is part of the widely documented shade avoidance syndrome of plants [2]. This phenotypic plastic response can be adaptive in the presence of competition for light, e.g., by elongating its stem and reaching sunlight and pollinators in a crowded ecosystem [4,5]. The ecological and adaptive significance and the physiological and genetic mechanisms underlying the phenotypic response of plants to shade are well documented [6,7]. However, little is known about the hypothesis that molecular epigenetic variation might underlie this ecologically relevant plastic response of natural populations (but see [8,9,10]).

### 1.1. Calling for Ecologically Relevant Tests of the Epigenetic Basis of Phenotypic Plasticity

Epigenetic changes can be involved with phenotypic plastic responses at the molecular level [11,12]. For example, the chromatin organisation and structure can change in relation to DNA methylation or histone post-translational modifications, which can affect gene expression and release transposable elements (TE) [13]. There is growing evidence for epigenetic variation associated with trait variation and phenotypic plasticity [14]. For example, phenotypic plasticity in response to temperature changes—either heat or cold treatments—was found to be associated with epigenetic modifications [15,16]. The role of epigenetic variation as an interface between ecological and genetic mechanisms is increasingly put forward in evolutionary biology studies [17]. More empirical work is needed to assess the ecological significance of epigenetic variation to understand its role in the evolution of natural populations. It is therefore necessary to test whether ecologically relevant phenotypic plastic responses are associated with epigenetic changes. Here we tested the hypothesis that the phenotypic plasticity in response to shade observed in snapdragon plants [1,8] was associated with epigenetic modifications by using an epigenomic approach.

### 1.2. Separating Genetic and Epigenetic Effects

Ecological and evolutionary epigenetics is a young domain of research that is constantly ongoing technical developments. One issue with epigenetic approaches of phenotypic variation is that the effect of DNA methylation changes can only be assessed in the absence of confounded genetic variation. This constraint challenges the use of epigenetics in studies at the scale of populations. Although statistical approaches are available to estimate simultaneously the genetic and epigenetic variation of phenotypic traits [18,19], they demand a quantity of data that is not adapted for small experiments on epigenomic variation. We, therefore, chose to use a technical solution to this issue. We used highly inbred lines of snapdragon plants in which genomes are nearly if not totally fixed in a homozygous state by successive generations of self-fertilization. We submitted plants from each snapdragon line to regular sunlight or shade, which allowed us to exclude or extremely reduce confounded genetic effects within lines and replicate the experiment across genetically different backgrounds.

### 1.3. Snapdragon Plants: The Road So Far

Previous work using High Performance Liquid Chromatography suggested that global methylation contents might change under different light treatments, and called for investigating DNA methylation patterns at the genomic level [8]. Several approaches can be used to characterize DNA methylation, such as Whole-Genome Bisulfite Sequencing (WGBS), Bisulfite converted restriction site associated DNA sequencing or bsRADseq, Epi RADseq, methylated DNA Immuno Precipitation, or meDIP. We chose epiGenome Bisulfite Sequencing or epiGBS [20]. This approach characterizes a reduced representation of the genome and therefore aims at detecting global patterns of DNA methylation changes spread across the genome. It is not aimed at identifying a specific gene or genomic region. Although the use of epigenomic methods is still restricted to small sample sizes, which impedes the study of multiple populations, the epiGBS approach allowed us to study enough samples to estimate the effect of ecological factors in snapdragon plant inbred lineages.

We aimed to assess whether the phenotypic response of snapdragon plants to light versus shade was associated with changes in DNA methylation patterns at the genomic level. We first assessed phenotypic differences associated with light versus shade by exposing plants grown in experimental to regular sunlight or artificially generated shade. Second, we tested whether the light versus shade treatment had an effect on global methylation patterns across the genome by sampling regions of the genome. Finally, we tested whether DNA methylation changes were consistently associated with phenotypic differences when we had found a significant effect of the light treatment on DNA methylation patterns.

## 2. Materials and Methods

### 2.1. Study System

We used four inbred lines of *Antirrhinum majus* L. (snapdragon plants) that were produced following successive generations of self-fertilisation. Snapdragon plants harbour hermaphroditic flowers that are usually self-incompatible [21]. It is a short-lived perennial plant characterized by zygomorphic flowers with genomic development and selection that is widely documented [22]. Its natural populations are highly genetically diverse [23,24] and geographically distributed across a large range of environmental conditions, in particular in terms of vegetation cover [25]. Snapdragon plants are locally adapted to their abiotic environment [26], and have been shown to react in terms of growth and development to light quality and intensity [1,27,28]. It is therefore an ecologically relevant study system to investigate the epigenomic basis of phenotypic plasticity in response to shade. Phenotypic plasticity in response to shade was already shown in experiments based on natural populations [1]. Here we chose to study highly inbred lines of snapdragon plants to exclude confounding genetic effects. We used lines from different origins to replicate our approach in different genetic backgrounds. These lines were originally made for horticultural and developmental genomics research programs. Three of them were provided by the John Innes Centre (Norwich Research Park), namely Ji75, Ji98, and Si50. The fourth line, namely E165, was obtained from the Technical University of Cartagena (Instituto de Biotecnología Vegetal, Pr Marcos Egea Gutiérrez-Cortines).

### 2.2. Experiment

The plant experiment was conducted outside under semi-controlled environmental conditions in the experimental garden facility of the ENSFEA agronomic school of Castanet-Tolosan, France (see photo in the Appendix A). Seeds were sown on 23 April 2018 in racks filled with mixture compost (50% BP2 Kompact 294, 50% TS3 Argile 404; Klasmann, Bourgoin Jallieu, France). Soon after germination when all seedlings harboured their first two or four leaves (4 June 2018), seedlings were transplanted in individual 9 × 9 cm pots filled with the same mixture compost. Every pot included one plant and was randomly assigned a location in the experimental garden. Half of the plants were randomly chosen and exposed to a shade treatment by using individual shading cages covered with net producing 70% shade. Plants were watered manually with the same amount of water twice a week. A total of 200 plants (50 per inbred line) were used in this experiment. In each inbred line, 25 out of the 50 plants were exposed to shade. The impact of shade nets on light intensity was characterized in a previous study by using multiple spectrophotometer acquisitions. They let pass through around one third of the photosynthetic active radiation (PAR) and two-thirds of the red to far-red ratio (R/FR) [1].

### 2.3. Adult Plant Phenotypic Measurements

One month after the young seedlings were exposed to shade (2 July 2018), adult plants were measured to test for the effect of shade on phenotypic measurements. Phenotypic measurements included height in cm, number of branches, presence or absence of floral buds, number of internodes, and stem diameter in mm. Internode length was calculated as the average stem length in cm per internode (plant height/number of internodes). Five fully developed leaves were collected, scanned, dried (3 days at 45 °C), and weighed. The area of leaves was measured using the ImageJ software [29] and the specific leaf area (SLA) was calculated as leaves surface (m^2^)/leaves mass (kg).

### 2.4. Second Round of Stem Growth for Plants Sampled for Tissue

After the first round of measurements presented above, stems of adult plants were cut at the first internode level to allow the growth of a new stem, still under the same light or shade treatment. This allowed us to sample tissue from meristems that were young enough not to be close to the stage of producing terminal flowers but for plants that had been exposed to shade or light for more than a month. Shoot apices were chosen because it is the place where new tissues start their differentiated growth and development. This is also where plants perceive external signals that drive phenotypic responses linked to growth or development. The shoot apex and two leaves were harvested on 48 plants, representing six plants by line and by treatment for each type of tissue. We also recorded phenotypic measurements to allow their comparison between treatments to be directly related to epigenetic data. These plants were measured on four different dates to allow their comparisons at the same developmental stage rather than age: three to four developed internodes (on 25 and 30 July and 1 and 6 August respectively 23, 28, 30, and 37 days after cutting). The same phenotypic measurements were taken as for adult plants during the first round of measurements, with an exception made for the presence or absence of a floral bud (none were present). The six phenotypic traits that were analysed were therefore: plant height, internode length, stem diameter, number of leaves, number of branches, and SLA.

### 2.5. Epigenetic Analysis

Tissue samples were frozen in liquid nitrogen at the moment of sampling and conserved at −80 °C until epigenetic analyses. We chose to sample shoot apexes because it is the part of the plant wherein all new tissues start their growth and develop. When an external signal is perceived by a plant and transformed into a phenotypic response that will drive the modification of the main stem and the organs located onto it (e.g., leaves), the perception and initiation of the response are expected to take place in the stem apical meristem. We also chose to sample leaves in order to explore the epigenetic variation expected to be associated with SLA plasticity. We chose to collect these two tissues because methylation was previously shown to be tissue-specific and so may vary differently between tissues responding to the same environmental treatment [30,31].

Shoot apices and leaves were ground to powder by using Tissue Lyser II (Qiagen, Hilden, Germany) which disrupts biological samples through high-speed shaking in plastic tubes with stainless steels. Total DNA was extracted by using Biosprint 15 DNA Plant kit (Qiagen, Hilden, Germany) which is a rapid and economical automated method that allowed purification of total DNA from plant tissue.

DNA methylation was studied by using the epiGBS method as the *Antirrhinum majus* reference genome was not available at the time [20]. In a nutshell, epiGBS is a reference-free reduced representation bisulfite sequencing method. This method uses genotyping by sequencing of bisulfite-converted DNA followed by reliable de novo reference construction, mapping, variant calling, and distinction of single-nucleotide polymorphisms (SNPs) versus methylation variation (protocol details can be found in the Appendix A). All library preparations have been realized by Niels Wagemaker (department of Experimental Plant Ecology, Radboud University, Nijmegen) according to their published protocol [20].

### 2.6. Bioinformatics

We used a bioinformatics pipeline integrated in the snake-make workflow called epiGBS2 [32] to remove PCR duplicates and demultiplex samples. The pipeline is available at: https://github.com/nioo-knaw/epiGBS2. The filtered and demultiplexed reads from epiGBS2 pipeline were used in another pipeline adapted from previous work [33], using the Galaxy project server as applied in [34]. Adapter removing was done using TrimGalore! V06.5 [35]. Single-end reads were aligned to the snapdragon plant genome version 3.0 [36] with BSMAP Mapper [37]. Mapped reads were merged and used as input in BSMAP Methylation Caller to get a tabular file with cytosine and thymine counts that were used as input to calculate coverage and Frequency of C and T for subsequent analysis.

CpG methylated sites with coverage of at least eight reads per position found in all samples were filtered with the package Methylkit [38]. After BSMAP methylation calling, bedgraph files were used to filter the sites in a CHG and CHH methylation context where H can be A, C, or T. Only the methylation sites covered by eight or more reads were retained for the Principal Component Analysis.

### 2.7. Statistical Analysis

We summarized the epiGBS data on DNA methylation changes by using PCA for each methylation protocol and line. PCA was conducted with the package factoextra, FactoMineR, emmeans, and missMDA (scripts available at the end of the Appendix A). We retrieved PCA coordinates per individual, the relative and absolute contributions of the components (also named dimensions) to the global variance, and the contribution of variables (cytosines positions) to the components (also named dimensions) by DNA methylation context (CpG, CHG, and CHH) and by tissue (apex and leaves) for the subsequent analyses.

We assessed the effect of the light versus shade treatment on (1) the phenotypic traits (plant height, internodes length, stem diameter, number of flowers, number of ramifications, and SLA), and (2) methylation patterns summarised by PCA dimensions with Mann–Whitney U-tests for each line, tissue, and methylation protocol. We extracted the effect size and its 95% confidence interval for each test, which allowed us to assess and compare the effect of the light versus shade treatment among lines, tissues, methylation protocols, and PCA dimensions. Where methylation differences due to the light treatment were found, we assessed the correlation between phenotypic trait values and variation in the methylation patterns represented by PCA dimensions with a Spearman correlation test. Each test was conducted on 12 individuals and replicated in the four snapdragon plant inbred lines, 11 to 12 PCA dimensions, two tissues (apex and leaves), and three methylation protocols (CHG, CHH, and CpG). Effect size estimates with confidence intervals that did not include zero were considered as significantly different from zero. 

We estimated effect sizes and their 95% confidence interval from Mann–Whitney tests using the R packages “rcompanion” [39] and “coin” [40]. We estimated the Spearman correlation coefficients and their 95% confidence interval using the R package “RVAideMemoire”. All analyses were performed in R software version 3.6.3 [41].

We performed power analyses of the Mann–Whitney tests depending on the effect size *r* and power analyses of the correlations depending on the Spearman correlation coefficient *r_S_* (see ‘power analysis’ in the Appendix A for more details and Appendix A).

## 3. Results

### 3.1. Phenotypic Response to Light Versus Shade Treatment

Our analysis revealed a strong effect (effect size *r* > 0.5) of the light versus shade treatment on all phenotypic traits considered except for height (Figure 1, Appendix A). Snapdragon plants exposed to regular natural light had more branches, shorter internodes, a larger basal stem diameter, more leaves, and were shorter than their counterparts exposed to shade. We found no difference between inbred lines in the strength of their response to the light treatment, as illustrated by nearly complete overlap between the 95% CIs of the light treatment effect between lines (Appendix A). One must note that we had limited statistical power to detect the significance of small size effects (Appendix A).

### 3.2. PCA Summary of DNA Methylation Data

The different PCA dimensions explained very similar percentages of DNA methylation data variation for each PCA across the 11 dimensions summarizing the CHH and CHG data variation, and across the 12 dimensions summarizing the CpG data variation, both for apex and leaf tissue (Appendix A). Caution must be taken when interpreting the 12th dimension because it explained only c. 10 to the minus 29 power % of the variation. Since DNA methylation data variation could not be summarized to a very low number of dimensions (Appendix A), we kept all the dimensions of each PCA and considered them equivalent in the statistical analyses used to test for associations between phenotypic traits measurements and DNA methylation changes.

### 3.3. DNA Methylation Association with Light Versus Shade Treatment and Phenotypic Variation

The analysis of the effect of the light versus shade treatment on DNA methylation revealed variation within lines between light and shade treatments. Caution must be taken when considering this variation and one should not speculate as to its statistical significance because 95% CI generally overlapped the zero. However, one to three PCA dimensions reflected a large difference between light and shade treatments (r ≥ 0.5) that can be unambiguously considered as significant in apex tissue (Figure 2, Appendix A).

Equivalent results were found in leaf tissue (Figure 3, Appendix A).

### 3.4. Association between Phenotypic Differences and DNA Methylation Changes

Among the 11 cases presented above that showed wide and significant methylation pattern differences associated with the light versus shade treatment, phenotypic variation was not always found to correlate with DNA methylation variation represented by PCA coordinates. Interestingly, no significant correlation was found between height and PCA coordinates in apex tissue (Figure 4A, Appendix A). In the analysis based on leaf tissue samples, it was the number of leaves that did not show any link with DNA methylation variation (Figure 4B, Appendix A). Caution must be taken when interpreting these relationships as they characterize the correlation between trait and epigenetic variations but do not take into account the treatment effect.

## 4. Discussion

### 4.1. Phenotypic Plasticity

Our results confirmed snapdragon plant phenotypic plasticity to light versus shade in highly inbred lines. It is interesting to note that this finding on phenotypic plasticity was replicated in a similar experimental setting for wild snapdragon plant populations [1]. Such phenotypic plastic response is typical of the response described in the presence of shade avoidance syndrome [2,3]. For a similar height, snapdragon plants exposed to shade had flatter or thinner leaves (increased SLA), which is usually associated with a higher growth rate in favorable environments. They were also characterized by increased stem elongation, which is an increase in the mean internode distance and one if not the most documented example of plasticity in plants. Plants grew bigger under shade as illustrated by their higher number of branches, larger basal stem diameter, and greater number of leaves. Although increased internode length and SLA are commonly reported in response to shade, branching is usually reduced because of apical dominance [2]. Our results, together with results found in myrtle plants (*Myrtus communis*) are starting a pool of examples of branching increased by shade [42]. Collectively, these findings and previous findings in wild snapdragon populations and inbred lines support the hypothesis of a strong phenotypic plasticity in response to shade in snapdragon plants. They also suggest that this plasticity was conserved in snapdragon horticultural lines. Interestingly, the magnitude of the plastic response was comparable between lines.

### 4.2. Epigenetic Response to Light Versus Shade

Previous studies on snapdragon plants found that their ~400 to 500 Mb genome largely harbored methylations to a non-negligible extent (15%), which is comparable to several plant species and seems to vary in relation to the genome size in Angiosperms [8,43]. Our results in highly inbred lines showed that methylation patterns on the snapdragon genome can change in response to the modification of the light environment (sunlight versus shade) of the plants. This result was found in different highly inbred lines that have fully or nearly fixed genomic backgrounds. It connects indirectly epigenetic variation to the ecology of natural populations in different genomic backgrounds. Equivalent examples of this biological link can be found in the literature [44,45]. Here we call for more studies in non-model organisms. This will be necessary before we can obtain a clear picture of the ecology and evolution of genetic and epigenetic variation at the level of populations [46].

Our results suggest that DNA methylation variation was spread across the genome because the different dimensions of the PCA that summarized the variation of DNA methylation patterns across the genome provided a balanced explanation of the variation. Although the coordinates of most PCA dimensions varied between the light versus shade environments, only a few of these dimensions underwent a strong significant effect: the others were only indicative of trends in a low statistical power context. We therefore cannot conclude to the presence of global epigenetic response to shade across the genome. Instead, our results suggest that a limited number of epigenomic regions were involved in a strong modification of DNA methylation patterns in response to light versus shade environments. One could speculate about the interest of precisely identifying these regions but since the epiGBS approach covers a small percentage of the genome, whole-genome approaches would be more suitable for this aim. This limitation is inherent to reduced representation sequencing methods. Our findings, therefore, imply some strong but regionally restricted epigenomic changes in snapdragon plants in response to light versus shade environments. Epigenetic variation in snapdragon plants therefore participates to the schoolbook example of ecologically and evolutionary relevant examples of phenotypic plasticity in plants.

To date, our study is one of the very few that investigates the potential link between the phenotypic plasticity of plants in response to shade and epigenetic variation. For example, clonal lines of longstalk starwort plants (*Stellaria longipes*) submitted to different light treatments showed that stem elongation correlated with reduced methylated cytosine content measured by High Performance Liquid Chromatography (HPLC) [9]. In *Arabidopsis thaliana*, histone acetylation of H3/H4 and H3K4me3/H3K36me3 promoted the expression of shade responsive genes in the Col-0 genotype [10]. Collectively, these findings and ours suggest that beyond the widely documented genetic mechanisms underlying the phenotypic plasticity of plants in response to shade [4,7], epigenetic variation might potentially be involved. Further work is necessary before any finding can be generalized to other plant species.

### 4.3. The Epigenetic Basis of Phenotypic Variation or Lack Thereof

Although we found that phenotypic variation was often associated with variation in DNA methylation patterns in different highly inbred lines, no clear relationship between trait phenotypic plasticity and epigenetic change emerged. We found an epigenetic basis for trait variation, and its absence, in many scenarios. For example, it was the case for traits that did not change in response to shade, for traits that changed although no epigenetic modification was found, when neither the trait nor DNA methylation patterns were modified by the light versus shade treatment, but also when both responded. Our analysis was inconclusive and neither confirmed nor denied the hypothesis that epigenetic modifications played a role in snapdragon trait phenotypic plasticity in response to shade.

A clear pattern of DNA methylation variation among snapdragon lines emerged from the analysis of trait epigenetic variation. This finding confirms at the epigenomic level the results found previously by using chemical analyses (8); there are differences in the epigenetic variation of traits between genomic backgrounds. This is highly expected because DNA methylation variation is linked to the DNA sequence. DNA sequence polymorphism at potentially methylated cytosine sites results in methylation variation [47]. Other mechanisms link DNA sequence polymorphism to methylation patterns, e.g., the mobility of Transposable Elements (TEs) enabled by changes in DNA methylation [48] and epigenetically facilitated DNA mutation [17]. Our study illustrates that methodological developments are still necessary in non-model species to overcome limits in the study of the ecological and evolutionary significance of epigenetic variation.

## 5. Conclusions

Our findings and others suggest that epigenetic variation might be associated with the phenotypic plasticity of plants in response to shade. This plasticity is likely influencing the ability of most plant populations to adapt. Beyond its general ecological relevance in nature, it has implications for the ongoing challenges linked to climate change. This is because contemporary changes of the vegetation cover can be observed in many ecosystems worldwide because of fragmentation [49] and land-use changes [50]. Plant vegetative architecture and photosynthetic related traits also play a key role in the evolution, adaptation, and plasticity of crop plants (crop breeding and domestication [51,52,53,54]). We therefore call for more work on the potential epigenetic variation associated with the phenotypic plasticity of plants in response to shade because it would improve our understanding of the potential ecological and evolutionary significance of epigenetic variation in natural populations.

## Figures and Tables

**Figure 1 genes-12-00227-f001:**
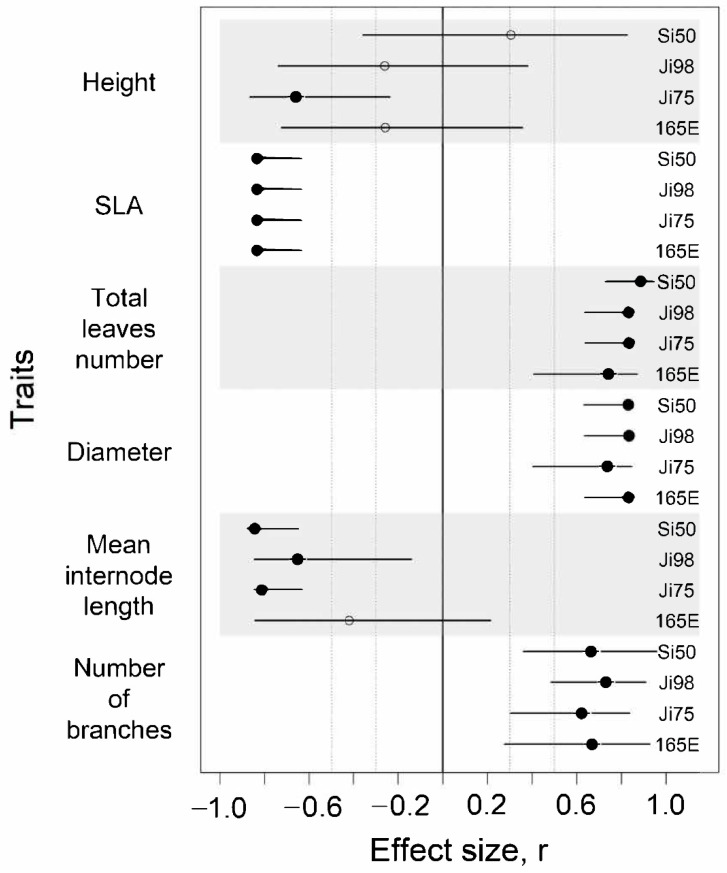
Effect sizes of light treatment on phenotypic traits represented for each snapdragon inbred line flanked by their 95% confidence interval. Line identities are noted in the column on the right side of the graph. Dotted lines represent *r* = 0.3 and *r* = 0.5. Estimates for which the 95% CI does not include zero are represented by black circles while others are represented by empty circles.

**Figure 2 genes-12-00227-f002:**
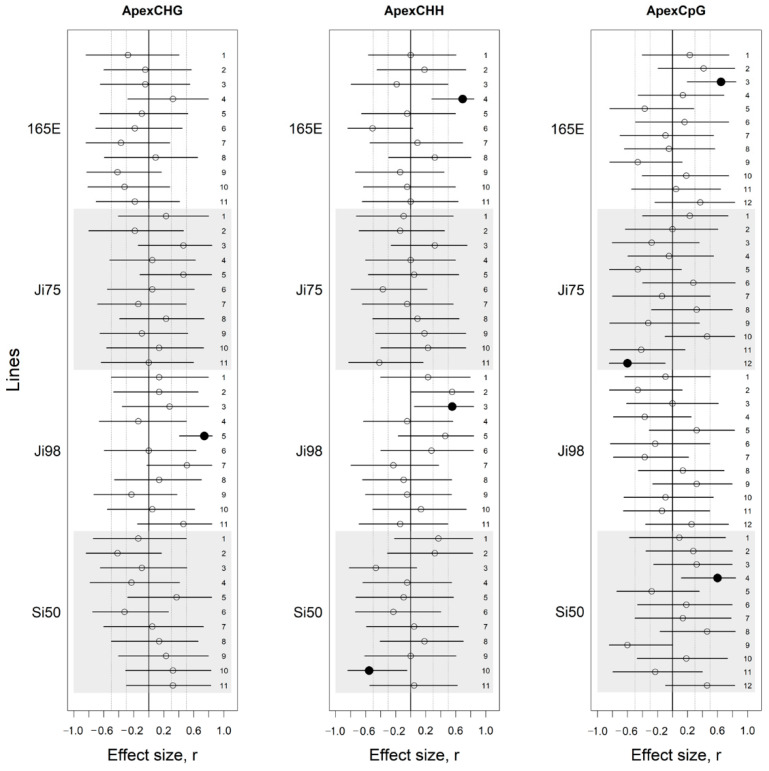
Effect sizes of methylation differences between light versus shade treatments presented for each methylation protocol (CHG, CHH, CpG) applied on apex tissue. Effect sizes are presented for each line and flanked by their 95% confidence interval. Dotted lines represent *r* = 0.3 and *r* = 0.5. Estimates for which the 95% CI does not include zero are represented by black circles while others are empty. Numbers on the right column show the PCA dimension of the methylation protocol.

**Figure 3 genes-12-00227-f003:**
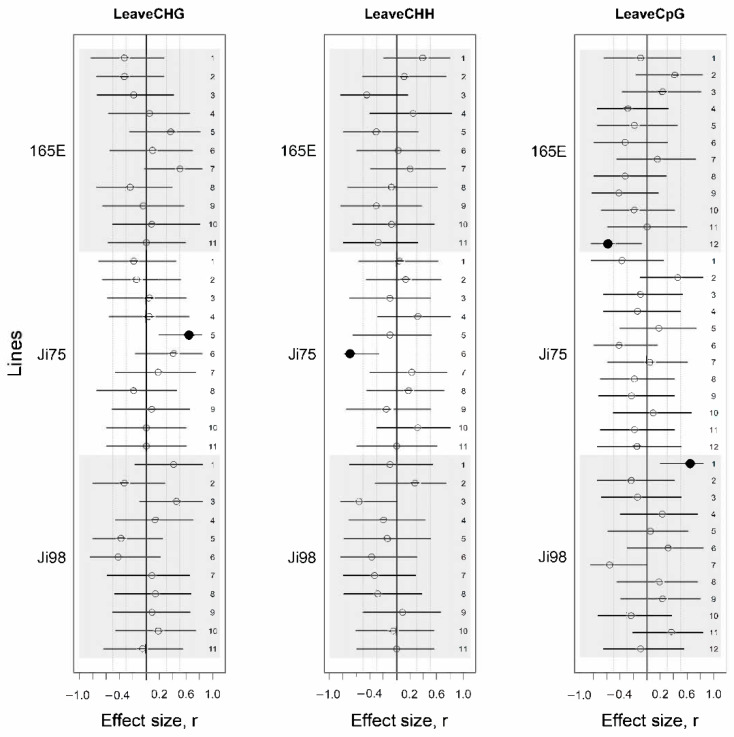
Effect sizes of methylation differences between light versus shade treatments presented for each methylation protocol (CHG, CHH, CpG) applied on leaf tissue. Effect sizes are presented for each line and flanked by their 95% confidence interval. Dotted lines represent *r* = 0.3 and *r* = 0.5. Estimates for which the 95% CI does not include zero are represented by black circles while others are empty. Numbers on the right column show the PCA dimension of the methylation protocol.

**Figure 4 genes-12-00227-f004:**
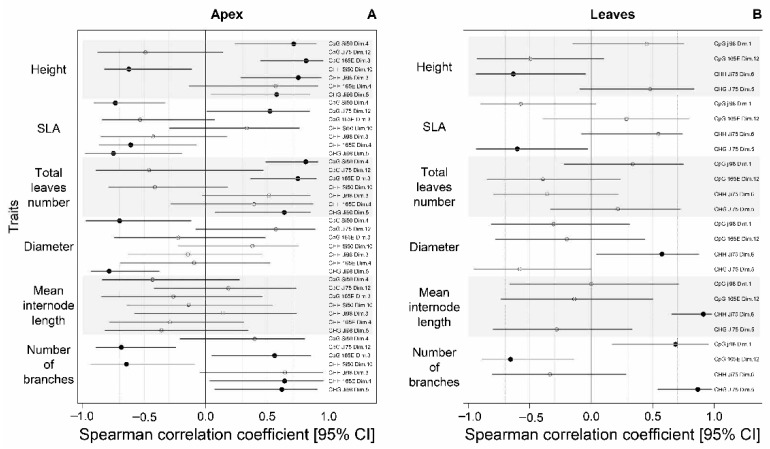
Correlations between each phenotypic trait and methylation PCA coordinate presented only for cases where DNA methylation differences due to the light versus shade treatment were found in apical (**A**) and leaf tissue (**B**). Spearman correlation coefficients (*r_s_*) presented are flanked by their 95% confidence interval. Dotted lines represent *r_s_* = 0.3, *r_s_* = 0.5 and *r_s_* = 0.7. Estimates for which the 95% CI does not include zero are represented by black circles while others are empty. Right column shows the methylation protocol, the line, and the PCA dimension for which the correlation coefficient is presented.

## Data Availability

Data will be made available upon request.

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
