# Peer review of "Phenotypic Response to Light Versus Shade Associated with DNA Methylation Changes in Snapdragon Plants (Antirrhinum majus)"

_genes, 2021, doi:10.3390/genes12020227_

Round 1

Reviewer 1 Report

In this MS, the authors analyzed phenotypic variation and DNA methylation changes in four lines under different environments: light and shade. It is interesting to link phenotypic plasticity and DNA methylation. However, some issues authors need to figure out before accepting.

In the "Materials and Methods", the authors wrote, "We used lines from different origins to replicate our approach in different genetic backgrounds". Do the lines have a different response to the shade environment?  

In the "Materials and Methods" and 2.2. Experiment, authors wrote "A total of 200 plants (100 per inbred line) were used in this experiment. In each inbred line, 25 out of the 50 plants were exposed to shade. ", however, 4 lines were used in this experiment. Do authors want to say 50 per bred line? please check it.

In results 3.1, the authors wrote "Our analysis revealed a strong effect (effect size r > 0.5) of the light versus shade treatment on all phenotypic traits considered except for height (Figure 1, Table S1). Snapdragon plants exposed to regular natural light had more branches, shorter internodes, a larger basal stem diameter, more leaves and were shorter than their counterparts exposed to shade. In addition, inbred lines did not show any differences in their response to light treatment.". Do authors feel this statement with contradiction? Or authors mean between inbred lines did not show any differences? authors may need to reorganize the writing.

In results 3.2 authors wrote "Caution must be taken when interpreting the 12th dimension because it explained only c. 10-29% of the variation.", what does the "c. 10-29%" mean? typos?

Author Response

REF1

Open Review

English language and style

( ) Extensive editing of English language and style required
( ) Moderate English changes required
( ) English language and style are fine/minor spell check required
(x) I don't feel qualified to judge about the English language and style

Yes

Can be improved

Must be improved

Not applicable

Does the introduction provide sufficient background and include all relevant references?

( )

(x)

( )

( )

Is the research design appropriate?

( )

(x)

( )

( )

Are the methods adequately described?

( )

(x)

( )

( )

Are the results clearly presented?

( )

(x)

( )

( )

Are the conclusions supported by the results?

( )

(x)

( )

( )

Comments and Suggestions for Authors

In this MS, the authors analyzed phenotypic variation and DNA methylation changes in four lines under different environments: light and shade. It is interesting to link phenotypic plasticity and DNA methylation. However, some issues authors need to figure out before accepting.

In the "Materials and Methods", the authors wrote, "We used lines from different origins to replicate our approach in different genetic backgrounds". Do the lines have a different response to the shade environment?  

Authors’ response: They had a similar phenotypic response. This is illustrated by treatment effect 95% Confidence Interval which is nearly completely overlapping between lines. A sentence was added in the result section to emphasize that point: “We found no difference between inbred lines in their response to the light treatment, as illustrated by nearly complete overlap between the 95% CIs of the light treatment effect between lines (Table S1).”

In the "Materials and Methods" and 2.2. Experiment, authors wrote "A total of 200 plants (100 per inbred line) were used in this experiment. In each inbred line, 25 out of the 50 plants were exposed to shade. ", however, 4 lines were used in this experiment. Do authors want to say 50 per bred line? please check it.

Authors’ response: Well spotted, it is indeed 50 per line. we corrected that mistake

In results 3.1, the authors wrote "Our analysis revealed a strong effect (effect size r > 0.5) of the light versus shade treatment on all phenotypic traits considered except for height (Figure 1, Table S1). Snapdragon plants exposed to regular natural light had more branches, shorter internodes, a larger basal stem diameter, more leaves and were shorter than their counterparts exposed to shade. In addition, inbred lines did not show any differences in their response to light treatment.". Do authors feel this statement with contradiction? Or authors mean between inbred lines did not show any differences? authors may need to reorganize the writing.

Authors’ response: We clarified the wording by modifying the last sentence. Indeed, as mentioned by the reviewer, we meant that there was no difference between lines. The sentence now reads: “We found no difference between inbred lines in their strong response to the light treatment, as illustrated by nearly complete overlap between the 95% CIs of the light treatment effect between lines (Table S1).”

In results 3.2 authors wrote "Caution must be taken when interpreting the 12th dimension because it explained only c. 10-29% of the variation.", what does the "c. 10-29%" mean? typos?

Authors’ response: There was some formatting issue. It should have read c. “10-29”. We modified the text to avoid any potential issue. It now reads : “approximately 10 to the minus 29 power %”. It is a negligible amount of variation which why we call for caution when interpreting this axis because it does not mean much.

Submission Date

30 December 2020

Date of this review

18 Jan 2021 06:42:04

Reviewer 2 Report

The manuscript "Phenotypic response to light versus shade associated with DNA methylation changes in Snapdragon plants (Antirrhinum majus)" seems to be very interesting to understand if the genome methylation is the basis of the phenotypic changing of the plant due to light shades. 

Globally, the work is fine. The manuscript is clear and well written, especially from the materials and methods and on. I suggest changing partially the form of some long phrases in the introduction to facilitate the reading. I have listed some correction dived respect the paragraphs.

Abstract

  • Change "The phenotypic plasticity of plants in response to change in their light environment, and in particularly to shade, is a school book example of ecologically relevant phenotypic plasticity with evolutionary adaptive implications." in "The phenotypic plasticity of plants in response to change in their light environment, and in particular, to shade is a school book example of ecologically relevant phenotypic plasticity with evolutionary adaptive implications."
  • Change " Our approach contributed this new finding but illustrates the limits in terms of sample size and statistical power of population epigenetic approaches in non-model organisms." in "Our approach contributed to this new finding but illustrates the limits in terms of sample size and statistical power of population epigenetic approaches in non-model organisms.

Introduction

  • Change "Although the ecological and adaptive significance and the physiological and genetic mechanisms underlying the phenotypic
    response of plants to shade are well documented [6,7], little is known about the hypothesis that molecular epigenetic variation might underlie this ecologically relevant plastic response of natural populations (but see [8-10])." in "The ecological and adaptive significance and the physiological and genetic mechanisms underlying the phenotypic
    response of plants to shade are well documented [6,7]. Although this, little is known about the hypothesis that the molecular epigenetic variation might underlie this ecologically relevant plastic response of natural populations (but see [8-10])."
  • Change "Although the role of epigenetic variation as an interface between ecological and genetic mechanisms is increasingly put forward in evolutionary biology studies [17], more empirical work is needed to assess the ecological significance of epigenetic variation, and ultimately understand how its role in the evolution of natural populations. It is therefore necessary to test whether ecologically relevant phenotypic plastic responses are associated with epigenetic changes." in "The role of epigenetic variation as an interface between ecological and genetic mechanisms is increasingly put forward in evolutionary biology studies [17]. More empirical work is needed to assess the ecological significance of epigenetic variation to understand how its role in the evolution of natural populations. It is, therefore, necessary to test whether ecologically relevant phenotypic plastic responses are associated with epigenetic changes."
  • Change "We therefore chose to use a technical solution to this issue." in "We, therefore, chose to use a technical solution to this issue."
  • Change "Previous work using High Performance Liquid Chromatography suggested that global methylation contents might change under different light treatments, and called for investigating DNA methylation patterns at the genomic level [8]. Several approaches can be used to characterize DNA methylation; Whole-Genome Bisulfite Sequencing (WGBS), Bisulfite converted restriction site associated DNA sequencing or bsRADseq, Epi RADseq, methylated DNA Immuno Precipitation or meDIP, but we chose epiGenome Bisulfite Sequencing or epiGBS [20]." in "Previous work using High-Performance Liquid Chromatography suggested that global methylation contents might change under different light treatments, and called for investigating DNA methylation patterns at the genomic level [8]. Several approaches can be used to characterize DNA methylation, such as Whole-Genome Bisulfite Sequencing (WGBS), Bisulfite converted restriction site-associated DNA sequencing or bsRADseq, Epi RADseq, methylated DNA Immuno Precipitation or meDIP. We chose epiGenome Bisulfite Sequencing or epiGBS [20]."
  • Change "Our aim was to assess whether the phenotypic response of snapdragon plants to light versus shade was associated with changes in DNA methylation patterns at the genomic level." in "We aimed to assess whether the phenotypic response of snapdragon plants to light versus shade was associated with changes in DNA methylation patterns at the genomic level."

Material and Methods

  • Change "It is a short-lived perennial plant characterized by zygomorphic flowers which genomic development and selection is widely documented [22]." in "It is a short-lived perennial plant, characterized by zygomorphic flowers which genomic development and selection, is widely documented [22]."
  • Change "Here we chose to study highly inbred lines of snapdragon plants
    in order to exclude confounding genetic effects." in "Here we chose to study highly inbred lines of snapdragon plants to exclude confounding genetic effects."
  • Change "They let pass though around one third of the photosynthetic
    active radiation (PAR) and two third the red to far red ratio (R/FR) [1]." in "They let pass though around one-third of the photosynthetic
    active radiation (PAR) and two-third the red to the far-red ratio (R/FR) [1]."
  • Change "We used a bioinformatics pipeline integrated in the snakemake workflow called epiGBS2 [32] to remove PCR duplicates and demultiplex samples, the pipeline is available at: https://github.com/nioo-knaw/epiGBS2. The filtered and demultiplexed reads from epiGBS2 pipeline were used in another pipeline adapted from a previously work [33], using Galaxy project server as applied in [34]." in "We used a bioinformatics pipeline integrated into the snake-make workflow called epiGBS2 [32] to remove PCR duplicates and demultiplex samples. The pipeline is available at https://github.com/nioo-knaw/epiGBS2. The filtered and demultiplexed reads from epiGBS2 pipeline were used in another pipeline adapted from previous work [33], using the Galaxy project server as applied in [34]."

Results 

  • Change "In leaf tissue, it was the number of leaves that did not show any link with DNA methylation variation (Figures 4B, table S6)." in "In leaf tissue, it was the number of leaves that did not show any link with DNA methylation variation (Figures 4B, Table S6)."

Discussion

  • Change "It is interesting to note that this finding on phenotypic plasticity
    was replicated in a similar experimental setting for snapdragon plant wild populations [1] Such phenotypic plastic response is typical of the response described in the presence of a shade avoidance syndrome [2,3]." in "It is interesting to note that this finding on phenotypic plasticity was replicated in a similar experimental setting for snapdragon plant wild populations [1]. Such phenotypic plastic response is typical of the response described in the presence of a shade avoidance syndrome [2,3]."
  • Change "One could speculate about the interest to identify precisely these regions but since the epiGBS approach covers a small percentage of the genome, whole genome approaches would be more suitable to this aim. This limitation is inherent to reduced representation sequencing methods. Our findings therefore imply some strong but regionally
    restricted epigenomic changes in snapdragon plants in response to light versus shade environments." in "One could speculate about the interest to identify precisely these regions but since the epiGBS approach covers a small percentage of the genome, whole-genome approaches would be more suitable for this aim. This limitation is inherent to reduced representation sequencing methods. Our findings, therefore, imply some strong but regionally restricted epigenomic changes in snapdragon plants in response to light versus shade environments."

Author Response

REF2

Open Review

English language and style

( ) Extensive editing of English language and style required
(x) Moderate English changes required
( ) English language and style are fine/minor spell check required
( ) I don't feel qualified to judge about the English language and style

Yes

Can be improved

Must be improved

Not applicable

Does the introduction provide sufficient background and include all relevant references?

(x)

( )

( )

( )

Is the research design appropriate?

(x)

( )

( )

( )

Are the methods adequately described?

(x)

( )

( )

( )

Are the results clearly presented?

(x)

( )

( )

( )

Are the conclusions supported by the results?

(x)

( )

( )

( )

Comments and Suggestions for Authors

The manuscript "Phenotypic response to light versus shade associated with DNA methylation changes in Snapdragon plants (Antirrhinum majus)" seems to be very interesting to understand if the genome methylation is the basis of the phenotypic changing of the plant due to light shades. 

Globally, the work is fine. The manuscript is clear and well written, especially from the materials and methods and on. I suggest changing partially the form of some long phrases in the introduction to facilitate the reading. I have listed some correction dived respect the paragraphs.

Abstract

  • Change "The phenotypic plasticity of plants in response to change in their light environment, and in particularly to shade, is a school book example of ecologically relevant phenotypic plasticity with evolutionary adaptive implications." in "The phenotypic plasticity of plants in response to change in their light environment, and in particular, to shade is a school book example of ecologically relevant phenotypic plasticity with evolutionary adaptive implications."

Authors’ response: Done

  • Change " Our approach contributed this new finding but illustrates the limits in terms of sample size and statistical power of population epigenetic approaches in non-model organisms." in "Our approach contributed to this new finding but illustrates the limits in terms of sample size and statistical power of population epigenetic approaches in non-model organisms.

Authors’ response: Done

Introduction

  • Change "Although the ecological and adaptive significance and the physiological and genetic mechanisms underlying the phenotypic
    response of plants to shade are well documented [6,7], little is known about the hypothesis that molecular epigenetic variation might underlie this ecologically relevant plastic response of natural populations (but see [8-10])." in "The ecological and adaptive significance and the physiological and genetic mechanisms underlying the phenotypic
    response of plants to shade are well documented [6,7]. Although this, little is known about the hypothesis that the molecular epigenetic variation might underlie this ecologically relevant plastic response of natural populations (but see [8-10])."

Authors’ response: Done, we also replaced “although this” by “however”

  • Change "Although the role of epigenetic variation as an interface between ecological and genetic mechanisms is increasingly put forward in evolutionary biology studies [17], more empirical work is needed to assess the ecological significance of epigenetic variation, and ultimately understand how its role in the evolution of natural populations. It is therefore necessary to test whether ecologically relevant phenotypic plastic responses are associated with epigenetic changes." in "The role of epigenetic variation as an interface between ecological and genetic mechanisms is increasingly put forward in evolutionary biology studies [17]. More empirical work is needed to assess the ecological significance of epigenetic variation to understand how its role in the evolution of natural populations. It is, therefore, necessary to test whether ecologically relevant phenotypic plastic responses are associated with epigenetic changes."

Authors’ response: Done

  • Change "We therefore chose to use a technical solution to this issue." in "We, therefore, chose to use a technical solution to this issue."

Authors’ response: Done

  • Change "Previous work using High Performance Liquid Chromatography suggested that global methylation contents might change under different light treatments, and called for investigating DNA methylation patterns at the genomic level [8]. Several approaches can be used to characterize DNA methylation; Whole-Genome Bisulfite Sequencing (WGBS), Bisulfite converted restriction site associated DNA sequencing or bsRADseq, Epi RADseq, methylated DNA Immuno Precipitation or meDIP, but we chose epiGenome Bisulfite Sequencing or epiGBS [20]." in "Previous work using High-Performance Liquid Chromatography suggested that global methylation contents might change under different light treatments, and called for investigating DNA methylation patterns at the genomic level [8]. Several approaches can be used to characterize DNA methylation, such as Whole-Genome Bisulfite Sequencing (WGBS), Bisulfite converted restriction site-associated DNA sequencing or bsRADseq, Epi RADseq, methylated DNA Immuno Precipitation or meDIP. We chose epiGenome Bisulfite Sequencing or epiGBS [20]."

Authors’ response: Done

  • Change "Our aim was to assess whether the phenotypic response of snapdragon plants to light versus shade was associated with changes in DNA methylation patterns at the genomic level." in "We aimed to assess whether the phenotypic response of snapdragon plants to light versus shade was associated with changes in DNA methylation patterns at the genomic level."

Authors’ response: Done

Material and Methods

  • Change "It is a short-lived perennial plant characterized by zygomorphic flowers which genomic development and selection is widely documented [22]." in "It is a short-lived perennial plant, characterized by zygomorphic flowers which genomic development and selection, is widely documented [22]."

Authors’ response: Done

  • Change "Here we chose to study highly inbred lines of snapdragon plants
    in order to exclude confounding genetic effects." in "Here we chose to study highly inbred lines of snapdragon plants to exclude confounding genetic effects."

Authors’ response: Done

  • Change "They let pass though around one third of the photosynthetic
    active radiation (PAR) and two third the red to far red ratio (R/FR) [1]." in "They let pass though around one-third of the photosynthetic
    active radiation (PAR) and two-third the red to the far-red ratio (R/FR) [1]."

Authors’ response: Done

  • Change "We used a bioinformatics pipeline integrated in the snakemake workflow called epiGBS2 [32] to remove PCR duplicates and demultiplex samples, the pipeline is available at: https://github.com/nioo-knaw/epiGBS2. The filtered and demultiplexed reads from epiGBS2 pipeline were used in another pipeline adapted from a previously work [33], using Galaxy project server as applied in [34]." in "We used a bioinformatics pipeline integrated into the snake-make workflow called epiGBS2 [32] to remove PCR duplicates and demultiplex samples. The pipeline is available at https://github.com/nioo-knaw/epiGBS2. The filtered and demultiplexed reads from epiGBS2 pipeline were used in another pipeline adapted from previous work [33], using the Galaxy project server as applied in [34]."

Authors’ response: Done

Results 

  • Change "In leaf tissue, it was the number of leaves that did not show any link with DNA methylation variation (Figures 4B, table S6)." in "In leaf tissue, it was the number of leaves that did not show any link with DNA methylation variation (Figures 4B, Table S6)."

Authors’ response. There was no modification suggested in the reviewer’s comment. However, we improved this sentence to avoid any issue. It now reads: “In the analysis based on leaf tissue samples, it was the number of leaves that did not show any link with DNA methylation variation (Figure 4B, table S6). “

Discussion

  • Change "It is interesting to note that this finding on phenotypic plasticity
    was replicated in a similar experimental setting for snapdragon plant wild populations [1] Such phenotypic plastic response is typical of the response described in the presence of a shade avoidance syndrome [2,3]." in "It is interesting to note that this finding on phenotypic plasticity was replicated in a similar experimental setting for snapdragon plant wild populations [1]. Such phenotypic plastic response is typical of the response described in the presence of a shade avoidance syndrome [2,3]."

Authors’ response: Done

  • Change "One could speculate about the interest to identify precisely these regions but since the epiGBS approach covers a small percentage of the genome, whole genome approaches would be more suitable to this aim. This limitation is inherent to reduced representation sequencing methods. Our findings therefore imply some strong but regionally
    restricted epigenomic changes in snapdragon plants in response to light versus shade environments." in "One could speculate about the interest to identify precisely these regions but since the epiGBS approach covers a small percentage of the genome, whole-genome approaches would be more suitable for this aim. This limitation is inherent to reduced representation sequencing methods. Our findings, therefore, imply some strong but regionally restricted epigenomic changes in snapdragon plants in response to light versus shade environments."

Authors’ response: Done

Submission Date

30 December 2020

Date of this review

20 Jan 2021 15:18:26
